# Photoplethysmography for the Assessment of Arterial Stiffness

**DOI:** 10.3390/s23249882

**Published:** 2023-12-17

**Authors:** Parmis Karimpour, James M. May, Panicos A. Kyriacou

**Affiliations:** Research Centre for Biomedical Engineering, City, University of London, London EC1V 0HB, UK; parmis.karimpour@city.ac.uk (P.K.); james.may.1@city.ac.uk (J.M.M.)

**Keywords:** vascular ageing, arterial stiffness, photoplethysmography, PPG, cardiovascular disease, CVD

## Abstract

This review outlines the latest methods and innovations for assessing arterial stiffness, along with their respective advantages and disadvantages. Furthermore, we present compelling evidence indicating a recent growth in research focused on assessing arterial stiffness using photoplethysmography (PPG) and propose PPG as a potential tool for assessing vascular ageing in the future. Blood vessels deteriorate with age, losing elasticity and forming deposits. This raises the likelihood of developing cardiovascular disease (CVD), widely reported as the global leading cause of death. The ageing process induces structural modifications in the vascular system, such as increased arterial stiffness, which can cause various volumetric, mechanical, and haemodynamic alterations. Numerous techniques have been investigated to assess arterial stiffness, some of which are currently used in commercial medical devices and some, such as PPG, of which still remain in the research space.

## 1. Introduction

The primary aim of this review is to cover the current state-of-the-art technology and approaches in detecting arterial stiffness and monitoring vascular ageing. The review aims to evaluate current trends, especially Photoplethysmography (PPG)-based approaches, and to offer suggestions for future research. The concept of using non-invasive measuring techniques to evaluate vascular stiffness is presented along with the benefits and drawbacks of the available modalities.

Globally, cardiovascular disease (CVD) is the leading cause of mortality, accounting for approximately 17.9 million fatalities in 2019 [1]. As vessels age, the heart and blood vessels undergo changes, leading to degeneration and an increased risk of CVD. The arterial system’s ability to function may be harmed or even rendered ineffective by diseases that might develop as a result of vascular ageing. The diseases can impact upper and lower vasculature. Around 20% of people in the United Kingdom (UK) aged between 55 and 75 suffer from peripheral arterial disease (PAD), which is a narrowing of the artery vessels due to factors such as wall thickening or the accumulation of fatty deposits [2,3]. PAD patients typically have no outward symptoms, making early detection difficult. In extreme cases, this may necessitate even an amputation because in the absence of treatment, cardiovascular death may occur. Cardiovascular death can occur due to a lack of blood reaching the organs/tissues when the arteries become narrow or obstructed, a state known as gangrene. Affected limbs may have decreased perfusion and hence suffer from oxygen starvation, immobility, and increased discomfort [4]. Early detection can prevent cardiovascular death, and is crucial for asymptomatic patients. Advancements in technology and techniques to monitor vascular health can provide important information to avoid limb loss and other severe pathologies.

An artery wall is a muscular tube lined with smooth tissue consisting of three main layers: the intima, media, and adventitia [5]. Vascular ageing refers to the alteration of the mechanical and structural properties of the arterial wall with ageing. Collagen and elastin are the two fibres that make up the artery walls. Collagen replaces the elastin fibres as the vessels age, and the resulting collagen bridges cause the vessels to lose some of their elasticity. High blood pressure (BP), hypertension, can be brought on by arterial wall damage that restricts blood flow. Heart attacks, coronary artery disease (CAD), and arrhythmias are just a few of the many disorders that can develop as vessels age. As mentioned before, PAD, hypertension, and atherosclerosis (a build-up of plaque that obstructs an artery and impairs peripheral circulation) resulting in tissue damage and rest pain can develop. With so many diseases linked to the ageing of the vessels, it is crucial to evaluate vascular ageing effectively and at the early stages if possible. Since changes to the arteries’ elastic properties lead to an increase in arterial stiffness [6], the evaluation of vascular ageing relies heavily on the measurement of arterial stiffness. Figure 1 illustrates a comparison between a healthy and unhealthy vessel in terms of elasticity. 

Non-invasive imaging techniques such as ultrasound and Magnetic Resonance Imaging (MRI) are common in clinical practice. Angiography, a more invasive technique, in which a contrast agent is injected into the vessel [8], is also utilised. These techniques, while routinely used, are perhaps not ideal when rapid and early screening might be the desirable way forward. Such techniques can be cumbersome, expensive, operator-dependent, and sometimes invasive. In addition, patients must be referred by a general practitioner (GP) for hospital-based screening, which might delay detection and increase patient concern. The use of non-invasive sensing modalities for the assessment of vascular disease has attracted a lot of attention from researchers throughout the years. Despite many efforts, a quick and relatively simple screening method that is clinically acceptable is not yet available.

Photoplethysmography (PPG) is an optical technology that has risen in popularity. It is suggested that PPG can be a solution when developing novel techniques for assessing vascular ageing. Arterial stiffness can cause haemodynamic, mechanical, and volumetric changes. In aged vessels, the unhealthy vessels become stiff and thicken, becoming less elastic and making it more difficult for blood to pass through. Blood flow is altered in ways that can damage downstream organs. A higher pulse pressure results from higher systolic arterial pressure and lower diastolic arterial pressure. Unhealthy vessels experience volumetric changes as a result of having less pulsatile expansion than healthy vessels. These changes can influence the PPG signal. Therefore, PPG-based technology has the potential to evaluate viscoelastic properties of arteries, including monitoring haemodynamic and volumetric blood changes, and hence evaluate vascular ageing. 

The current methods utilised in clinical settings to evaluate vascular health have limitations. This literature review aims to explore state-of-the-art non-invasive techniques for assessing vascular ageing through measurements of arterial stiffness. By examining prior research conducted in in vitro and in in vivo settings, while emphasising key discoveries and outcomes along with the strengths and weaknesses of each technique, this review proposes recommendations for assessing vascular health in the future. While existing methods are inconvenient for GP-level screening, PPG has the potential to provide a non-invasive optical solution for assessing vascular ageing. PPG has gathered attention for its ability to assess viscoelastic properties, monitor haemodynamic changes, and detect volumetric alterations in blood flow. As a result, it offers promise in assessing vascular health by addressing the shortcomings of existing techniques. Nonetheless, further research is crucial for understanding the full scope of the capabilities and limitations of PPG before its widespread adoption in clinical practice.

## 2. Methods

Literature was identified through searches conducted in PubMed, Embase, Scopus, Web of Science, and Cochrane Library. Keywords such as “vascular ageing” and “non-invasive”, and “arterial stiffness”, were used in assessing the titles and abstracts. Several permutations of “photoplethysmography” were also employed. Only publications from 1990 to November 2023 were included to assess the shift from traditional to recent technologies. Papers published in the English language were only included in the search. This yielded a total of 543 papers from all the databases. Studies involving in vivo and in vitro experiments met the inclusion criteria. A total of 266 duplicate papers were eliminated. Furthermore, 213 papers were omitted because their titles and/or abstracts were unrelated to the study. Therefore, 64 papers were included for analysis by meeting the inclusion criteria (Figure 2). The searches were performed in November 2023. 

Out of the 64 studies reviewed, 32 evaluated arterial stiffness using PPG-based methods. Another 32 studies concentrated on alternative measurement techniques. Two PPG-based studies found in the search overlapped with another method; they have been included in the overall count for PPG-based procedures. The split for the mean number of publications found per decade can be seen in Figure 3. Between 2010 and 2019, demand rose for arterial stiffness measurement techniques, particularly those using PPG-based methods, compared to the period from 2000 to 2009. The search and inclusion criteria revealed a lack of interest in arterial stiffness assessment techniques from 1990 to 1999. 

## 3. Current State-of-the-Art Techniques and Technologies in Vascular Ageing

There are currently non-invasive methods and tools available to measure arterial stiffness. The calculation of indices, such as the augmentation index (AIx), ambulatory arterial stiffness index, and cardio-ankle vascular index (CAVI), and the measurement of Pulse Wave Velocity (PWV) are examples of current methodologies. Additionally, imaging methods such as ultrasound and MRI have been investigated over the years. Research has also been conducted utilising Laser Doppler Flowmetry (LDF) and near-infrared spectroscopy (NIRS). Researchers have examined the use of already-available instruments such as the Arteriograph device (TensioMed Ltd., Budapest, Hungary), the Mobil-O-Graph 24 h Pulse Wave Analysis (PWA) Monitor Device (I.E.M. GmbH, Stolberg, Germany), and more, including oscillometric devices, as well as computational approaches using algorithms and software programmes. More information on these subjects is covered later in the review, including their advantages and limitations.

### 3.1. The Recent Use of Measurements of Pulse Wave Velocity to Assess Arterial Stiffness

PWV is considered the gold standard for assessing arterial stiffness. PWV assesses the speed at which arterial pressure waves move through the aorta and large arteries. Recently, different techniques have been employed to analyse PWV in the assessment of arterial stiffness. Research has been conducted in areas linking arterial stiffness to body fat percentage (BFP), aortic valve replacement, children and adolescents with inflammatory bowel disease (IBD), and coronary stenosis. The use of impedance cardiography in assessing arterial stiffness using PWV has also been explored. 

Gong et al. [10] conducted a study to examine the relationship between BFP and arterial stiffness. Eligible patients were categorised into four groups based on gender and arterial stiffness, and carotid femoral PWV (cfPWV) was measured. Each patient’s BP, height, waist circumference, and body weight were also recorded. Given the relatively limited sample size, it was challenging to confirm a cause-and-effect relationship between BFP and cfPWV. PWV Tonometry-based devices were also utilised to measure the cfPWV prior to this investigation. Ji et al. [11] used cfPWV measurements to assess arterial stiffness. The study focused on the operator’s placement, the distance being measured, and the tonometer position. Manual measurements were made of the distances between the suprasternal notch (SNN) and the remote detection site (femoral artery), the distance between the sternal notch and the proximal (carotid artery) detection point, and the distance between the femoral artery and the carotid artery detection point (cf-distance). Western nations make extensive use of cfPWV, and the study’s tonometry-based apparatus is frequently used to assess aortic PWV. The device allowed for the measurement of central BP; however, questions over the accuracy of doing so with non-invasive methods remain. 

Cantürk et al. [12] investigated the severity of aortic stenosis after an aortic valve replacement. The patients were between the ages of 43 and 75. PWV was measured at baseline and six months after having an aortic valve replaced. However, a limited sample size was employed, and the study’s methodology was different from that of earlier research, making comparison challenging. A larger investigation is needed for more conclusive results. 

The arterial stiffness of children and adolescents with IBD has also been examined. Based on the carotid artery and the femoral artery, the Vicorder device (Skidmore Medical Limited, Bristol, UK) was used to take cfPWV measurements. A causal interpretation of the findings was constrained because it was a cross-sectional study of a relatively small and heterogeneous sample. Additionally, no comparison groups were available, meaning the outcomes had to be interpreted in relation to pre-determined reference values. Future research on the long-term effects of CVDs in patients with IBD should make use of bigger sample numbers [13].

Liu et al. [14] used multidetector computed tomography angiography to determine coronary stenosis using brachial-ankle PWV (baPWV). Although only elastic arterial stiffness should be recorded, one limitation of baPWV is that the parameter captures both muscle and elastic arterial stiffness. In another study, PWV was examined to determine whether it reflects central or peripheral arterial stiffness. However, one of the study’s shortcomings was that a longitudinal follow-up was not conducted to examine subsequent CVD occurrences. The results’ applicability to other populations is also uncertain despite the sample size being rather large [15]. In another study, Yufu et al. [16] measured the aortic PWA and percent mean pulse amplitude (%MPA). These tests were performed on 33 patients at the right brachial artery on both sides of the ankle. However, the study omitted the consideration of additional factors impacting %MPA beyond aortic stiffness, such as femoral stiffness.

Scudder et al. [17] recorded dual impedance signals using a spot electrode arrangement on 78 adults aged 19 to 78. The pulse transit time (PTT), which was obtained by measuring the aortic flow onset and arrival times of peripheral pulse waveforms, was used to calculate PWV. PTT has an inverse relationship to PWV and represents the amount of time it takes for a pulse wave to travel a known distance. The study had drawbacks despite the authors’ claim that d-ICG is a reliable approach to measure arterial stiffness, reasonably inexpensive, and easy to use due to minimal expertise being required. Firstly, the sample size was small and comprised, mostly, young healthy adults. Secondly, because of the longer pulse trajectory through distal arteries, such as the tibial artery, d-ICG was unable to provide direct assessments of central aortic stiffness (as with most non-invasive measures of PWV). Furthermore, it was advised that specific software configurations needed to be changed in order to collect the whole signal range and use the first derivative to determine timing characteristics.

### 3.2. The Recent Use of Arterial Stiffness Index Calculations to Assess Arterial Stiffness

The use of indices for measuring arterial stiffness has been studied by numerous researchers. This section summarises the recent utilisation of different indices, including the CAVI, ambulatory arterial stiffness index, AIx, arterial velocity–pulse index (AVI), arterial pressure–volume index (API), compliance index (CI), arterial stiffness index (ASI), and ankle-brachial index (ABI).

In order to determine arterial stiffness, Miyoshi and Ito [18] used the CAVI, which measured the distance from the origin at the aorta to the ankle and calculated the time it took for the pulse pressure wave to travel from the aortic valve to the ankle in order to obtain the PWV from the heart to the ankle. Additionally, the brachial artery in the upper arm was used to gauge BP. Following that, the Bramwell–Hill formula was applied to express the link between the PWV and change in volume. The fact that the CAVI can analyse arterial characteristics by splitting them into BP and arterial stiffness is one of its advantages over other methods for measuring arterial stiffness. Additionally, it measures the ascending aorta, suggesting that the arterial stiffness measurement may be more closely related to cardiac function. The CAVI has the potential to be helpful in clinical settings because of its relative simplicity and flexibility, though more research is needed.

Souza-Neto et al. [19] evaluated the ambulatory arterial stiffness index for the measurement of arterial stiffness in heart transplant patients. It was determined that the ambulatory arterial stiffness index presented a non-invasive way to indicate hypertension. However, the accuracy was questioned due to the study’s small sample size. Moreover, the index was not compared with standard methods such as PWV or AIx. In rheumatoid arthritis (RA) patients, Klocke et al. [20] used radial PWA to determine AIx. The subjects were between the ages of 18 and 50, and the results were contrasted with that of a healthy control group. Despite this, validation is still needed from larger research that might examine further elements of the illness. Investigations into known physiological factors that can affect the AIx are also necessary. As an illustration, it was stated that AIx was lower in men and that it negatively correlated with height and heart rate and positively correlated with age and peripheral BP; however, these results must be investigated further.

Zhang et al. [21] measured AVI and API to evaluate early atherosclerosis. As stated in Table 1, baPWV was unable to predict the presence of early atherosclerosis. However, this could be explained as, according to Vlachopoulos et al. [22], baPWV is the average arterial stiffness between the brachial and ankle arteries, which reduces sensitivity in detecting changes in central arterial stiffness. Despite this, the study had some drawbacks, including a sample size with a high proportion of patients who had taken cardiovascular medications such as anti-hypertensive medications. The effects of such medications remain unknown [21]. Similarly, the study conducted by Wang et al. [23] comparing clinical features in patients with chronic kidney disease (CKD) using measurements of arterial stiffness in various arteries did not consider the effects of medications. To assess local vascular stiffness, CI was utilised to analyse the correlation between volume and pressure changes at the fingertip. There were 186 CKD patients and 46 healthy subjects. The study had limitations in terms of the types of cardiovascular risk factors and, as mentioned earlier, the medications examined; not all risk factors were considered. Furthermore, because it was a cross-sectional study, causality could not be established.

The ASI, which was used to quantify arterial distensibility, ABI, and the arterial wave pattern using an oscillometric automated digital BP instrument, was studied by Choy et al. [24]. The subjects were split into two groups: 266 newly diagnosed stroke patients, ranging in age from 26 to 98, and a control group of 629 volunteers, all of whom were older than 30. A greater ASI indicated a higher likelihood of stroke incidence. However, ASI was determined based on BP readings, which can change depending on the participant’s health at the time of the examination. Subjects who had previously experienced a transient ischemic attack or a mild stroke may not have been detected despite efforts to rule out past stroke experience based on a questionnaire. Additionally, the effects of medications on ASI were not investigated.

### 3.3. The Recent Use of Imaging Techniques to Assess Arterial Stiffness 

#### 3.3.1. Ultrasonography 

One of the key techniques for assessing vascular age is ultrasound, an acoustical imaging modality, which uses high-frequency sound waves. Researchers have employed ultrasound to assess arterial stiffness, yielding promising results. However, ultrasound methods tend to be expensive and require expertise to operate. 

Naessen et al. [25] conducted a study involving 30 healthy individuals (median age, 62; range, 27 to 82), 19 patients with pulmonary arterial hypertension (median age, 53; range, 27 to 84), and 14 patients with left ventricular heart failure with a reduced ejection fraction (median age, 67; range, 48 to 82). The healthy subjects were non-smokers and had no prior history of heart or arterial disorders. Additionally, the subjects were not taking any medications that would have impacted the arterial wall. The study disproved the common belief that vascular changes in pulmonary arterial hypertensions are only related to lung vasculature. 

Li et al. [26] utilised ultrasound imaging techniques to assess arterial stiffness in patients with acute ischemic stroke (AIS) using real-time shear wave elastography to measure longitudinal elasticity modulus. With this technology, 50,000 images could be captured every second. Furthermore, radio frequency ultrasonography technology was used to calculate the PWV of the bilateral carotid arteries. The results demonstrated that shear wave elastography can be used in vascular applications. Future research is necessary to evaluate risk factors and assign various weights to arterial stiffness in longitudinal and circumferential directions because the study was unable to analyse all deviations.

Age-dependent elasticity variations in the common carotid artery elasticity served as the foundation for the comparison between conventional and ultrasonographic strain measures in a study conducted by Bjällmark et al. [27]. The evaluation involved 10 younger subjects, between the ages of 25 and 28, and 10 older subjects, between the ages of 50 and 59. The conclusion drawn was that two-dimensional (2D) strain imaging exhibited greater accuracy compared to traditional measurements. However, concerns arose regarding the stiffness indices, as these were derived from BP and lumen diameter measurements obtained at different sites, casting doubt on the accuracy of the variables.

#### 3.3.2. Magnetic Resonance Imaging 

MRI, which uses magnetic fields to create images, has also been investigated by researchers for the assessment of arterial stiffness. 

The study conducted by Kang et al. [28] used cardiac MRI to calculate the pulmonary artery distensibility index to assess pulmonary artery stiffness. However, the study had certain limitations. Firstly, despite the disease being rare, the subject group was small. Secondly, among the thirty-five patients under observation, only three were noted to have severe pulmonary regurgitation (PR), which can suggest that the pulmonary artery distensibility index would have been overestimated.

The impact of age on turbulent blood flow was assessed by Ha et al. [29] using four-dimensional (4D) flow MRI. As stated in Table 1, all subjects experienced turbulent flow in the aorta. However, this was based on a small sample size of twenty healthy males aged between 67 and 74 and twenty-two healthy males aged between 20 and 26. Although MRI does not expose patients to radiation, some may experience claustrophobia, rendering it inappropriate for a general evaluation of vascular ageing.

### 3.4. Laser Doppler Flowmetry and Near-Infrared Spectroscopy

The medical technique of LDF uses the concept of Doppler shift, which is the change in the frequency of light (in this case, laser). Sorelli et al. [30] assessed the vascular ageing using the Periflux 5000 LDF system (Perimed, Järfälla, Sweden). On the right hallux pulp of the individuals, microvascular perfusion was recorded. A supervised classifier was trained and validated using over 20,935 models of pulse waves. Although using LDF is less expensive than using other imaging techniques, it can have signal processing and motion artefact issues [43].

NIRS is an analytical technique that uses a broad spectrum of near-infrared light to illuminate the region of interest and measures the light that is absorbed, transmitted, reflected, or scattered. Age-related metabolic and microvascular function changes were assessed by Rogers et al. [31] using NIRS vascular occlusion testing. NIRS signals were recorded from 17 younger and 17 middle-aged and older women. The Framingham risk calculator was used to determine a 10-year risk. Due to the cross-sectional nature of the study, it was not possible to evaluate the causal temporal relationship between age and the results. Additionally, the thickness of the adipose tissue was not assessed, which might have reduced the absolute NIRS signals. Finally, the melanin levels were not considered, which might have affected the NIRS signal.

### 3.5. The Use of Existing Devices to Assess Arterial Stiffness

Commercial devices which measure arterial stiffness do exist. These devices include the Mobil-O-Graph 24 h PWA Monitor Device, SphygmoCor-Px System (AtCor Medina, Sydney, Australia), VP-1000 system (Omron Healthcare, Hoffman Estates, IL, USA), EndoPAT 2000 system (Itamar Medical, Franklin, MA, USA), PeriScope (M/S Genesis Medical Systems, Hyderabad, India), Form PWV/ABI vascular diagnostic device (Omron Healthcare, Kyoto, Japan), SonoSite 180 Plus (SonoSite Inc., Washington, DC, USA), Anteriograph, and NICOM (Cheetah Medical, Portland, OR, USA).

The Mobil-O-Graph 24 h PWA Monitor Device was used by Silva et al. [32] to evaluate the link between body composition and arterial stiffness. Dual-energy X-ray absorptiometry on a Hologic bone densitometry machine (Model Discovery A, Waltham, MA, USA) was used to obtain the body compositions of the participants. However, the study was constrained because it was carried out in a single location with community elders, lacking applicability to a broad and diverse population. Additionally, the study ignored the individuals’ usage of medications such as anti-hypertensives, which would have affected the central circulation parameters.

Although there are instruments that can distinguish between healthy and unhealthy vessels, no device is ideal. Perrault et al. [33] compared the measurement capabilities of the SphygmoCor-Px System, the VP-1000 system, and the EndoPAT 2000 system in healthy subjects ranging in age from 23 to 71. The outputs of the instruments differed numerically, making it difficult to compare the outcomes. Furthermore, because there is a lack of knowledge regarding changes to the parameters in relation to illness progression, it makes it difficult to track disease progression or the efficacy of a particular intervention. Markakis et al. [34] evaluated the SphygmoCor-Px System and the Mobil-O-Graph 24 h PWA Monitor Device in terms of feasibility. Invasive and non-invasive measurements were performed within 24 h of admission and again 48 hours later for comparison purposes on patients experiencing hemodynamic shock in intensive care units (ICUs). It was concluded, as illustrated in Table 1, that non-invasive procedures can be used as part of an additional monitoring method while invasive techniques are more trustworthy. However, the results of the study are questioned because of the small sample size. The SphygmoCor-Px has also been used in studies to assess vascular ageing by acquiring the PWV. Costa et al. [35] used the SphygmoCor-Px in a study conducted on in arterial hypertension individuals. Chi-square analysis was performed as the primary statistical analysis. Despite the small size representing a community and the requirement for a larger size, the study was able to demonstrate its ability to measure arterial stiffness non-invasively using a gold-standard instrument.

Sridhar et al. [36] used the PeriScope to measure arterial stiffness by obtaining the PWV of 988 healthy controls and 2988 who had a high risk of developing CVD. Using an oscillometric approach, the PeriScope simultaneously assessed the cfPWV and baPWV. It was concluded that those with a higher risk of developing CVD had a higher PWV than the healthy control group. In another study, Komine et al. [37] used an oscillometric BP device to evaluate arterial stiffness. An inflatable cuff was used to assess the BP of the individuals and the Form PWV/ABI vascular diagnostic device was used to obtain the baPWV and cfPWV. The SonoSite 180 Plus was used in the study to obtain images using ultrasound. It should be noted that the study did have restrictions. For instance, the oscillometric cuff pressure was used to take an indirect measurement of arterial volume. Between the brachial blood vessel and cuff, the size of the muscle and fat has unknown effects on the arterial volume. To determine whether the technique could be utilised to measure arterial stiffness when illnesses are present, more research is needed. In a more recent study, Hoffmann et al. [38] used a oscillometric BP monitor to assess early vascular ageing biomarkers in cosmonauts, aged between 41 and 51 (7 males and 1 woman), undergoing long-term space flights. To summarise, it was found that the cosmonauts had not undergone any clinically significant changes despite comparisons to the baseline measurements (represented by 65 to 90 days prior to the flight). Long-term flights in deep space should reportedly be looked upon in the future. Due to the study’s limited sample size, it is recommended to conduct a follow-up examination several years later. The current follow-up duration in the study was deemed too short to adequately identify any delayed onset of vascular diseases. 

Juganaru et al. [39] collected arterial stiffness parameters from 184 males and 129 females, ranging in age from 18 to 53, using the Anteriograph instrument. According to the study, a high waist-to-hip ratio may be a vascular stiffness risk factor. The early asymptomatic detection of vascular atherosclerosis could be aided by screening healthy people with high waist-to-hip ratios. Despite the device’s non-invasiveness, quick deployment, and repeatability in assessing artery stiffness parameters, incorrect cuff positioning can produce false findings. Osman et al. [40] evaluated arterial stiffness changes in low-risk pregnant women using the Arteriograph. Non-invasive evaluations were performed on low-risk pregnant women. A cuff was placed on the right arm over the brachial artery to use the Arteriograph. The NICOM was also used in the study for measurements including those of cardiac output, stroke volume, and total peripheral resistance. Advantageously, the Arteriograph instrument has been extensively utilised in pregnancy research and has been validated in invasive and non-invasive measurements in non-pregnant populations. Additionally, to reduce bias in the results, only one expert who was trained on using both the Arteriograph and NICOM made the recordings. The authors were able to identify developments or changes in arterial stiffness over a longer length of time because the longitudinal investigation was based on five distinct occasions. This study was based on a small number of participants; however, a larger population is required in future studies. 

### 3.6. The Recent Use of Computational Algorithms to Assess Arterial Stiffness

Over the years, researchers have created various computational algorithms to predict arterial stiffness. Computational algorithms navigate around the challenges associated with gathering extensive in vivo datasets along with the associated time and costs [44]. Kostis et al. [41] predicted arterial stiffness from pulse pressure using an algorithm. To test whether arterial age predicted stroke better than chronological age, two indices of arterial stiffness were created by the algorithm and adjusted for specified demographics. The study was constrained, nevertheless, because the algorithms had been demographically adjusted and hence could not be applied to other datasets. As a result, the approach could not be used with different datasets that included, for instance, different age and gender groupings. Having said that, and according to the authors, the approach may be used to design and carry out new randomised clinical trials.

To assess local arterial stiffness, a semi-automatic vendor-independent software was created by Negoita et al. [42] using a Vivid E95 ultrasound (GE Healthcare, Illinois, US) to collect images. The edges of the luminal arterial walls (M-mode) and blood velocity were determined by the software, and diameter and velocity waveforms were extracted from the ultrasound images. The study was assessed on healthy volunteers aged between 22 and 32, with four of them being females. The technique was vendor-independent; therefore, it could be used to analyse ultrasound images of diameter and velocity recorded on any ultrasound machine as long as they have been saved in the Digital Imaging and Communications in Medicine (DICOM) format. 

## 4. Using Photoplethysmography for Arterial Stiffness Assessment

PPG is a widely used non-invasive optical technique. It aids in studying and monitoring pulsations associated with changes in blood volume in a peripheral vascular bed. Over the last thirty years, the number of published articles on PPG has significantly increased, covering both basic and applied research. Throughout these publications, PPG has been praised as a non-invasive, low-cost, and simple optical technique for measuring physiological parameters applied at the surface of the skin. 

The popularity of this topic can be attributed to the realisation that PPG has important implications for a wide range of applications. Amongst many, it aids in blood oxygen detection, cardiovascular assessment, and vital sign monitoring. In addition, the significant contribution of PPG in wearable devices has exponentially elevated the popularity and usability of PPG.

Currently, there exists a large body of literature that contributes new knowledge on the relationship between PPG pulse morphology, PWA, and pulse feature extraction with the physiological status of peripheral blood vessels. This encompasses aspects such as ageing, stiffness, BP and compliance, and microvascular disease, amongst others. There are also significant efforts in the utilisation of the PPG for the detection of heart arrhythmias such as Atrial Fibrillation (AF). Researchers are continuing to strive to combine the PPG sensory capabilities of wearables, such as smartwatches, with Artificial Intelligence (AI) in delivering ubiquitous health monitoring solutions that go beyond the current available heart rate wearables [45].

PPG sensors comprise Light Emitting Diode(s) (LEDs) and photodector(s). The emitted light, which is made to transverse the skin, is reflected, absorbed, and scattered in the tissue and blood. The modulated light level, which emerges, is measured using a suitable photodetector. For example, it is possible for the hand to be directly transilluminated where the light source, usually in the broad region of 450 nm to 960 nm, is on one side of the skin and the detector is on the other side. This method, also called the transmission mode, is limited to areas such as the finger, the ear lobe, or the toe. However, when light is directed into the skin, a proportion is backscattered, emerging near the light source. The light source and the photodetector can be positioned side by side. This method, also called the reflection mode, allows measurements on virtually any skin area. The intensity of reflected and backscattered light reaching the photodetector in either reflection or transmission mode is measured. The variations in the photodetector current are assumed to correlate to blood volume changes beneath the probe. These variations are electronically amplified and recorded as a voltage signal called the photoplethysmogram (Figure 4) [45]. 

The PPG signal can be impacted by a number of variables such as temperature variations, the measurement site, perfusion status, and motion artefacts. As mentioned earlier, the transmittance or reflectance mode of the PPG sensor can differ depending on the anatomical measurement site, as shown in Figure 5. Thus, the wavelength of the light source(s) must be accounted for depending on the mode and distance that the light must penetrate. For example, red and infrared light reaches deeper than green light. Peripheral vasoconstriction can cause low-quality signals while good skin contact has demonstrated high-quality signals [46,47].

Understanding the constraints of PPG monitoring holds significance. Since PPG detects light, it faces drawbacks such as susceptibility to interference from ambient sources, impacting measurement accuracy. Motion artifacts also pose challenges [48], though utilising post-processing algorithms can mitigate the interferences. Additionally, variations in skin tones can impact the signal as PPG relies on light–tissue interactions. Considering these factors is crucial when developing PPG-based sensors for experiments, whether in vivo or in vitro.

### 4.1. The Recent Use of Photoplethysmography in Studies to Assess Arterial Stiffness

The evaluation of arterial stiffness using PPG has recently become popular. Many researchers have attempted to understand arterial stiffness using PPG in in vivo settings. Some in vivo research focuses on specific conditions including pregnancy, obesity, and diseases such as CAD. Studies have also compared PPG-based devices to one another and to other modalities to assess the viability of using PPG-based devices. The topics will be discussed in further detail below.

### 4.2. Existing Photoplethysmography-Based Devices to Assess Arterial Stiffness

PPG is a well-established optical technology; many researchers have adopted or developed new PPG-based devices, such as through research based at university laboratories. Some researchers have conducted in vivo studies to distinguish between healthy and unhealthy patients bilaterally (such as between healthy and PAD subjects) or by obtaining pulse waveforms from one measuring site, such as the left index finger. This section will introduce novel and existing PPG-based devices in further detail. 

One of the earliest PPG research studies was conducted by Allen and Murray [49] on a group of healthy people. Firstly, PPGs were recorded from the right (R) and left (L) sides of six peripheral sites (that is, L and R ears, L and R thumbs, and L and R toes). To validate the electronic matching of right-to-left channels, a set of validation data was first gathered. Secondly, the healthy volunteers provided a set of physiological data derived from PPGs. These were the root mean square error (RMSE), which measured the differences between the right and left side, and cross-correlation analysis, which measured the degree of similarity. Allen and Murray [49] found that in healthy individuals, the right and left sides of the body were highly correlated, as perhaps expected. This work paved the way for further research that was conducted by Bentham et al. [50]. While it was already proven that in healthy individuals the PPGs from the right and left were highly correlated, Bentham et al. [50] obtained multi-site finger and toe PPG recordings from 43 healthy control patients and 31 PAD subjects to carry out another bilateral study. Beat-to-beat normalisation amplitude variability and pulse arrival time (PAT) were assessed in the frequency domain using magnitude-squared coherence (MSC) and in the time domain using two statistical techniques. When the results from the two subject groups were analysed, patients with PAD had a different signal on one side of the body compared to the other, unlike the healthy subjects. The work conducted by Bentham et al. [50] highlighted the possibility of distinguishing between healthy and PAD subjects. The clinical demographic dataset that was gathered for the study was nonetheless limited, and only a few fundamental variability variables were investigated in terms of PWA.

Brillante et al. [51] used PPG to non-invasively measure arterial stiffness from the left index finger in healthy people ranging in age from 18 to 67. The research focused on the impacts of categories such as age, gender, and race on different indices including stiffness index (SI) and reflection index (RI). Analyses based on simple correlation, Spearman’s correlation, and multivariate regression were performed. Although the study concluded comparisons between the different categories and the indices, the study’s sample of healthy adults over 65 was underrepresented. Similar to the study by Brillante et al. [51], where it was found that there were no differences between genders in terms of arterial stiffness measurements, the results obtained by Jannasz et al. [52] supported the notion that gender had no bearing on the likelihood of developing atherosclerosis. It should be emphasised, nevertheless, that the results were primarily focused on female participants, which may have influenced the findings. In another study conducted by Tąpolska et al. [53], the subjects were separated according to age, gender, and weight. The Pulse Trace PCA 2 device (Micro Medial, Rochester, UK) was used by placing a reader on the index finger to evaluate SI using PPG techniques. Although it was concluded that SI was more useful than RI, both can be used in clinical practice. 

The work carried out by Tanaka et al. [54] using PPG signals taken from an occluded finger has made it possible to take measurements of the small artery and arteriole in the future. The study used Bland–Altman plots to evaluate the degree of agreement between the finger arterial stiffness index (FSI) and finger arterial elasticity index (FEI). Regression analysis, linear analysis, and bi-logarithmic analysis were used. The work has provided confidence in measuring arterial stiffness in smaller vessels. 

Wowern et al. [55] conducted an in vivo trial on people of various ages and genders, including pregnant women, using the SphygmoCor-Px System and Meridian digital pulse wave analysis (DPA) (Salcor AB, Uppsala, Sweden). The experiment’s goal was to understand how repeatable the arterial stiffness parameters determined by DPA were. Measurements were obtained from the left index finger. Second derivatives of the wave reflections were used to analyse the PPG signal (obtained from the Meridian DPA), and Bland–Altman plots were employed for statistical analysis. Despite our suggesting that DPA is a valuable tool to gauge vascular health, it still must be further examined because none of the DPA variables produced optimum repeatability, making it impossible to rely solely on such methods. In another in vivo study, carried out by Djurić et al. [56], a PPG sensor was used to measure blood flow using scalar coefficients. The encouraging results obtained from the study in differentiating between different age groups (above and below 50 years) have shown that PPG could be used in the future for vascular ageing measurements. It should be noted that the results were preliminary and based on a limited number of samples. Furthermore, the study was based on healthy volunteers without including any patients with vascular diseases. Future work should consider more categories for the age groups, as well as the impact of vascular diseases. 

Huotari et al. [57] tested a transmission-probe-based PPG device created in a university lab. The sensor captured pulse waveforms, which were then mathematically decomposed to determine the arterial stiffness. However, because of the complexity of the hemodynamic features, it was challenging to calculate the arterial stiffness indices of arteries. Determining the link between PPG-derived indices and indices generated from pressure and flow pulses was a difficulty. The study demonstrated that arterial stiffness assessments using PPG had a promising future.

### 4.3. Use of Photoplethysmography on Specific Conditions

Researchers have implemented PPG-based techniques to assess arterial stiffness in patients with specific health conditions, such as CAD, heart transplantation, diabetes, hypertensive, and obesity, and high-risk patients. Pregnancy and cerebral pulsatility have also been investigated by researchers. The topics will be discussed in further detail. 

Zekavat et al. [58] evaluated the association of ASI with BP and CAD. Multivariable COX proportional hazards and additive linear regression were among the analysis models used. The results led to a lack of confidence that PPG-derived ASI could predict CAD risk, leaving a gap for future research. Arterial stiffness changes in heart transplant patients have been compared by Sharkey et al. [59]. The study was conducted on 20 children with heart transplantation and on a healthy control group of 161 children. Data were collected bilaterally from the ear lobes, index fingers, and great toes. The PPG signal collected from the children with heart transplants was normalised and compared to the normalised PPG signal from the control group. For statistical analysis, multivariate (that is, binary logistic regression (BLR)) and univariate analyses (that is, the Mann–Whitney U test) were performed. This study suggested the possibility to measure arterial stiffness at different body sites. Research has also been conducted on diabetic patients whereby there is a possibility to distinguish between diabetic and non-diabetic individuals using an arterial stiffness monitoring system based on PPG technology [60]. Furthermore, the second derivative of PPG has been investigated as a potential indicator of arterial stiffness. In a study conducted on 260 patients, it was found that the arterial stiffness progression differed in the diabetic and non-diabetic stages [61].

The second derivate of a PPG signal and the PWV were compared in a study by Bortolotto et al. [62] that examined vascular ageing evaluation in hypertensive participants. The study involved 524 patients with hypertension and 140 with atherosclerosis alteration, which included coronary heart disease, peripheral vascular disease, and abdominal aortic aneurysm. The second derivative of the PPG was suggested as a potential tool for assessing vascular ageing in hypertensives. The length of the vascular segment may have been overstated by the PWV approach, which should be considered despite the fact that PWV was a better indicator of the presence of atherosclerosis alteration than the second derivative of PPG.

Korneeva and Drapkina [63] investigated the possibility of using PPG on obese patients with high BP by assessing arterial stiffness. The main objective was to provide statins, namely atorvastatin and rosuvastatin, to these patients and track the development of vascular stiffness. In an additional effort, Drapkina and Ivashkin [64] used a PPG device attached to a finger to conduct a pulse wave study on arterial stiffness in obese and high-BP patients. Prior to this research, Drapkina [65] used a finger PPG device to examine arterial stiffness in high-risk patients with high BP. The study was carried out similarly to that by Korneeva and Drapkina [63], albeit for high-risk patients. Both sets of results supported one another, noting that high-risk or obese patients had increased arterial stiffness. 

Other circumstances, such as pregnancy-related circumstances, have also been analysed using PPG. The study conducted by Wowern et al. [66] involved PPG signals being collected from the left index fingers of healthy pregnant women. For analysis, linear and polynomial mixed effects were used to account for gestational age, and analysis of variance (ANOVA) and analysis of covariance (ANCOVA) were used to account for age influences. Yet, there was uncertainty and a lack of trust in the ability to identify pathological haemodynamic changes during pregnancy, thus postulating the necessity to investigate pathological changes that can occur during pregnancy. 

In two independent acute interventions (a cold pressure test and one involving mild lower-body negative pressure), Lefferts et al. [67] investigated the effects of cerebrovascular pulsatility in terms of acute increases in arterial stiffness in middle-aged and young adults. A mean BP reading was acquired using PPG for continuous BP monitoring. Cerebrovascular hemodynamics at rest, during the cold pressure test, and during the lower-body negative pressure intervention were evaluated in 15 middle-aged people between the ages of 47 and 61 and in 15 young adults with genders matched to the middle-aged. The measurements, however, were not evaluated constantly, resulting in a little variation in the timings of the measurements. The middle-aged adult group also included a small number of participants who had CVD risk factors, such as obesity and anti-hypertensive medication usage, which could have improved validity but affected the findings. Future research should involve older subjects, as the study only focused on those under 61 years of age. 

### 4.4. Comparing Current Photoplethysmography-Based Instruments and Other Measurement Methods

While some researchers have used custom-made PPG-based instruments for their work, others have compared several existing instruments. Complior (Alam Medical, Saint-Quentin-Fallavier, France), PulsePen (DiaTecne, Milan, Italy), and PulseTrace (GP Supplies, Borehamwood, UK) are the three devices that have been compared. PPG and measurement indices that gauge arterial stiffness, such the ABI, ASI, SI, and AIx, have also been examined. Additionally, imaging PPG (iPPG) and contact PPG (cPPG) have been compared. 

Salvi et al. [68] compared current commercial PPG devices, namely Complior, PulsePen, and PulseTrace. In contrast to Complior and PulsePen, which used aortic PWV based on the interval between carotid and femoral pressure waves, PulseTrace was evaluated using SI measures. There remains a need to standardise PWV measurements and establish a reference value by contrasting various devices. Djeldjli et al. [69] compared iPPG with cPPG. The experiment was based on healthy participants. Two probes were used to record the signals for the cPPG, one on the right earlobe and the other on the index finger. The study’s small sample size and primary focus on a certain age range and skin type limited the results to this application alone. Large-scale population testing is necessary to assess the impact of the measuring site on the measurements. 

Kock et al. [70] compared the ABI to PPG measurements for arterial stiffness in elderly patients. Many analysis models were used, such as bivariate and multivariate linear regression, the Shapiro–Wilk test to analyse residuals, and winsorisation. Measures of central tendency and data dispersion were used to denote quantitative variables, whereas absolute frequencies and percentages were used to represent qualitative variables. It was concluded that the ABI did not relate to PPG indicators, and more research is needed to establish standardised procedures for vascular assessment. In another study, using a fingertip PPG device, Murakami et al. [71] compared the ASI to the well-known baPWV. Through statistical analyses such as the non-parametric Wilcoxon rank sum test, Receiver Operating Characteristic (ROC) curve, and the Area under the Curves (AUCs), the instrument was shown to be capable of measuring arterial stiffness in accordance with baPWV. To ascertain the possibility of distinguishing between subjects with high and low cardiovascular risk, Clarenbach et al. [72] compared the SI acquired through PPG to the AIx obtained through radial tonometry. The study involved 62 individuals who had either chronic obstructive pulmonary disease (COPD) or obstructive sleep apnea, and 21 healthy volunteers served as controls. The subjects were between the ages of 18 and 75. Whilst both devices were successful, the SI acquired from the PPG device could further distinguish between intermediate and high-risk people, rendering it more effective in clinical settings. 

### 4.5. The Use of Photoplethysmography in Computational Models

Researchers have incorporated PPG into computational models to predict vascular health, as well as analysing incident and reflected waves from the PPG waveform. The advantages and disadvantages of using PPG in computational models are discussed. 

Machine learning (ML) and deep learning (DL) were applied to the PPG signals of a database in a study conducted by Dall’Olio et al. [73]. The approach involved pre-processing data on the raw PPG signals, through steps such as detrending, demodulating, and denoising. The DL employed the entire signal to predict healthy vascular ageing (by bypassing the feature extraction stage), whereas the ML relied on known extracted features from the PPG signal. For DL, it was possible to bypass the feature extraction stage as several convolutional neural networks (CNNs) were applied to the entire PPG signal as an input. Although CNNs with 12 hidden layers or fewer showed good performance, more complicated structures cannot be trained on a common laptop; subsequent research should explore more complex structures using a feasible technique. Due to the black-box approach being used, the study was restricted in its ability to compare its findings to those of other ML methodologies. In a study conducted by Shin et al. [74], DL was applied to the PPG pulses of individuals ranging in age from 20 to 80. DL offers an advantage over manually recognising features during the assessment of vascular ageing since it has the potential to produce features from PPG waveforms automatically. It is anticipated that more databases will become accessible in the future, resulting in the DL approach performing better in the assessment of vascular ageing. Nonetheless, the employment of computational models in wearable technology raises additional issues such as the computing power that may affect the ability to obtain an immediate diagnosis. Resolving computation model issues, such as determining the optimal number of hidden layers or the trade-off between the number of parameters and the volume of training data, is crucial [73]. 

Park and Shin [75] evaluated vascular ageing using an artificial-neural-network-based regression model to analyse incident and reflected waves from a PPG waveform. Their study report claimed that a trustworthy single PPG-based technique for assessing arterial stiffness had not yet been developed. The Gaussian mixture model was used to deconstruct each waveform into incident and reflected waves after the recorded PPG signals were segmented for each beat. Since the measurements were based on nasal PPG rather than the more typical finger PPG measurements, it was challenging to generalise the findings. It is unclear how this has an impact on the findings. Future research is needed to understand the effects of various measurement sites. Assessing the model’s performance in relation to risk factors that hasten vascular illnesses such as atherosclerosis, which were not considered in the study, is imperative.

In recent years, researchers have also shown interest in remote health monitoring, particularly through wearables [76]. This is driven by the ageing population, projected to increase by 10% in the next five years [77]. Remote PPG (rPPG), monitoring cardiovascular activity via facial video, has gained attention. Despite being labelled non-invasive and low-cost, rPPG faces challenges such as a low signal-to-noise (SNR) ratio. Lian et al. [78] proposed employing signal processing methods to enhance rPPG by reducing noise and improving accuracy. This involved data fusion, region of interest selection, and heart rate estimation. Furthermore, rPPG attracts attention for fatigue detection. Zhao et al. [79] utilised CNN for rPPG-based learning fatigue classification through multi-source feature fusion. However, the study used a self-collected dataset and lacked confirmation on enhancing fatigue detection accuracy with a larger sample size using DL. Previous research explored rPPG for estimating blood oxygen saturation (SpO_2_). Casalino et al. [80] credited rPPG as being portable, enabling continuous SpO_2_ monitoring. However, while rPPG is gaining popularity, it has not yet been applied rigorously in any large studies relating to vascular ageing. 

### 4.6. Combining Photoplethysmography with Other Modalities and Using Photoplethysmography to Assess Novel Developments

Some researchers have attempted to fuse PPG with other modalities, such as an electrocardiogram (ECG), to overcome some of the current drawbacks. In the same respect, the novel device known as the single continuous passive leg movement (sPLM) has been assessed as a possible screening technique using PPG technology. 

By combining multi-site PPG and ECG, Perpetuini et al. [81] performed an in vivo assessment of vascular stiffness. Ten ECG leads and eight PPG probes were used to collect signals. Signals could be simultaneously gathered from numerous places using various PPG probes. The ECG served as a reference for single-pulse PPG evaluation and averaging. Pressure cuffs were offered to ensure robust optode-to-skin connection. Given that the PPG–ECG system could record back-reflection signals at a significant inter-optode distance, it was hypothesised that PPG signals could be collected from large arteries. Additionally, numerous sites were monitored at once due to the large number of probes used. However, the results were not always interpretable when collecting multiple PPG signals, and significant computing resources may be required [82].

PPG was used in a study by Hydren et al. [83] in an effort to evaluate sPLM. All subjects were male and were split into two groups consisting of 12 younger and 12 older subjects. The instructions provided by Gifford and Richardson [84] for using the sPLM were followed. Although it is highlighted in Table 2 that a decline in vascular function brought on by ageing is sensitive to sPLM, further studies are required with larger samples prior to labelling the sPLM as a clinical tool for monitoring vascular ageing [83].

## 5. Discussion and Conclusions

The measurement and detection of vascular ageing is important, whether it be in a hospital setting, a GP clinic, or even a home setting. Early detection and guidance to individuals can offer patients time to change their lifestyles and to postpone the onset and progression of diseases such as atherosclerosis or PAD. Unfortunately, no device has yet been proven ideal. It is important that a future device should be relatively inexpensive, non-invasive, accurate, and user-friendly. As such, an easy-to-use measuring tool can allow for measurements to take place at home without the need to be referred by the GP to a clinical setting for a trained specialist to take readings, saving time and resources. 

Predominantly, imaging techniques used to assess vascular ageing are expensive, often impractical, and require a specialist. As stated before, procedures such as MRI and angiography must be carried out in a hospital setting where qualified professionals are on hand to take images and recordings. Despite the fact that these methods exist, patients still rely on GP referrals, which might delay the detection of vascular ageing and cause patients’ concern, especially in more remote areas where GP access might be difficult in the first place. Unfortunately, there is not a single imaging technique that can accommodate all patients; although MRI is inappropriate for claustrophobic patients, other methods can expose patients to radiation, and certain imaging modalities, such as angiography, are more invasive. Home monitoring is gaining popularity as wearable devices expand. Therefore, to continuously monitor vascular ageing in real time, it is crucial to have a practical, expert-independent, relatively inexpensive diagnostic instrument that can be integrated into smart homes or wearable technology. 

There are commercially available instruments to assess arterial stiffness, such as the SphygmoCor-Px System and the Mobil-O-Graph 24 h PWA Monitor Device; however, their use has certain limits. Nonetheless, potentially, there is a possibility of using commercially available tools at home to monitor progress on a frequent basis while, if necessary, maintaining the option to go to a clinical setting for more precise monitoring. 

Researchers are increasingly turning to PPG-based technology, which might provide an answer when developing and inventing novel devices to evaluate vascular ageing and arterial stiffness. Due to PPG’s simplicity of use, longitudinal studies could be performed to understand whether vascular ageing treatments can alter arterial stiffness and whether a PPG-based device can detect vascular ageing over time. PPG can easily be incorporated into devices and wearable technology, as has already been proven. PPG is suited for universal screening since it is easy to use, is relatively inexpensive, and does not need special training to operate. Another consideration, when creating novel technologies, should be the utilisation of a multi-sensor approach. Combining various sensors with PPG should be considered, as relying solely on PPG may not be the optimal solution. Employing a multimodal approach has the potential to address the challenges associated with PPG and result in a more reliable sensor technology for assessing vascular ageing.

To acquire more conclusive results from in vivo investigations, it is recommended that future studies use bigger sample numbers. Longitudinal studies should also be conducted to assess novel methodologies over the long term. Within study protocols, and as part of the recommended application of PPG devices, a strong contact pressure should be established because measurements can be affected by the quality of the contact [47]. Future research should focus on standardising measuring methods, expanding the databases accessible to computational models, and understanding the potential impacts of medications such as anti-hypertensive medications. Additionally, it is suggested that in the future, an in vitro system that can simulate the mechanical dynamics of a vascular system in a controlled environment be built. This way, pathologies can be introduced to imitate CVD disorders to verify the accuracy of novel devices.

In conclusion, this paper has reviewed and examined the current state-of-the-art techniques and technologies in the assessment of vascular ageing while highlighting the popularity of the well-established optical technique of PPG. Also, studies exploiting recently developed and commercially available devices have been discussed. In this review paper, the benefits and drawbacks of each study have been also discussed. Given that CVD is the main cause of mortality worldwide, it is only a matter of time before cost-effective, reliable screening based on PPG is needed. 

## Figures and Tables

**Figure 1 sensors-23-09882-f001:**
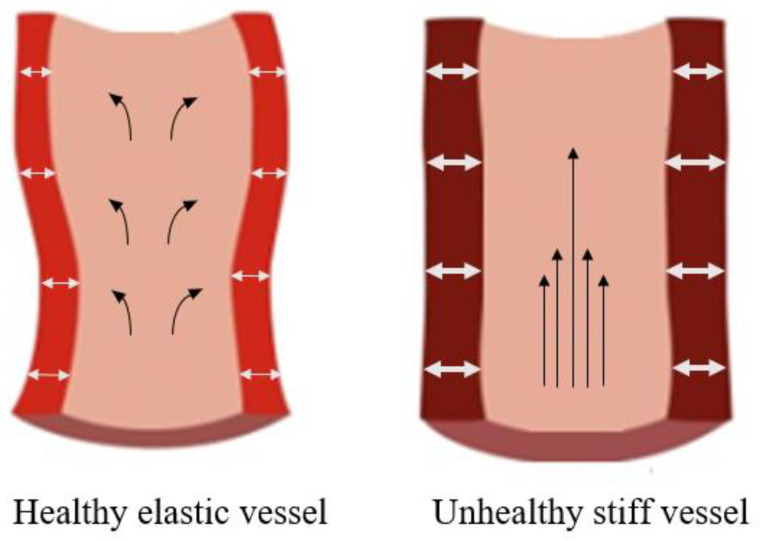
Comparison of healthy elastic vessel (**left image**) with an aged, unhealthy stiff vessel (**right image**). The unhealthy vessel comprises thicker walls and a uniform flow due to the stiffer walls. Meanwhile, the elasticity of the healthy vessel allows for an increased pulsatile flow. Modified from LaRocca et al. [7].

**Figure 2 sensors-23-09882-f002:**
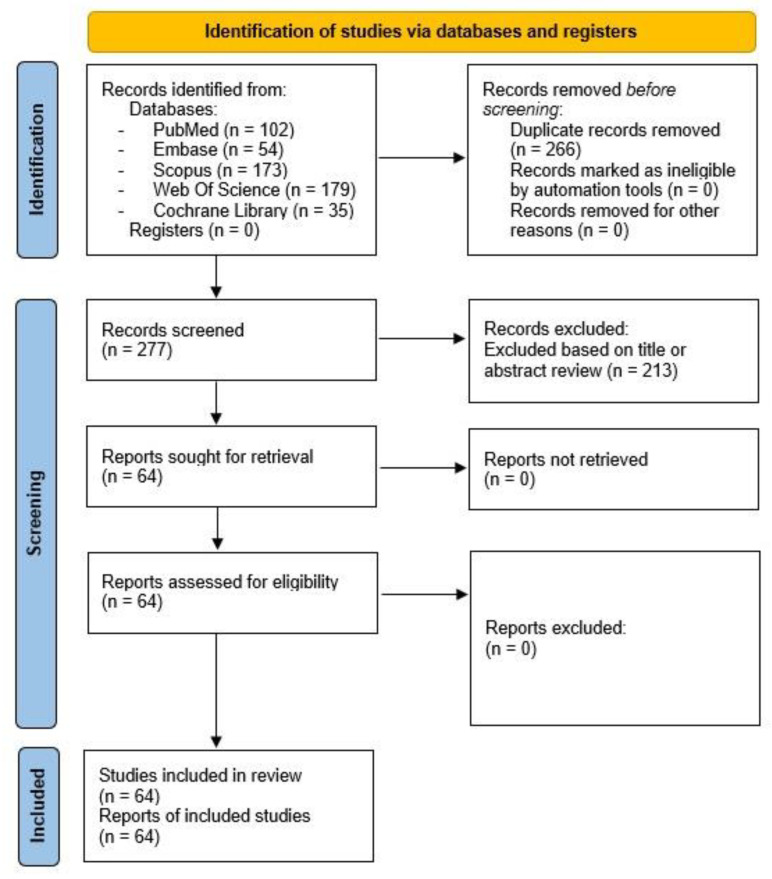
Flow diagram from PRISMA [9] illustrating the search strategy for the literature review procedure. The PRISMA flow diagram facilitated study selection. It was used to present the review clearly, mapping the number of papers and eliminating duplicates in databases. Records deemed irrelevant based on title or abstract were excluded from the screened total. Subsequently, 64 papers underwent eligibility assessment, all of which have been included in the review.

**Figure 3 sensors-23-09882-f003:**
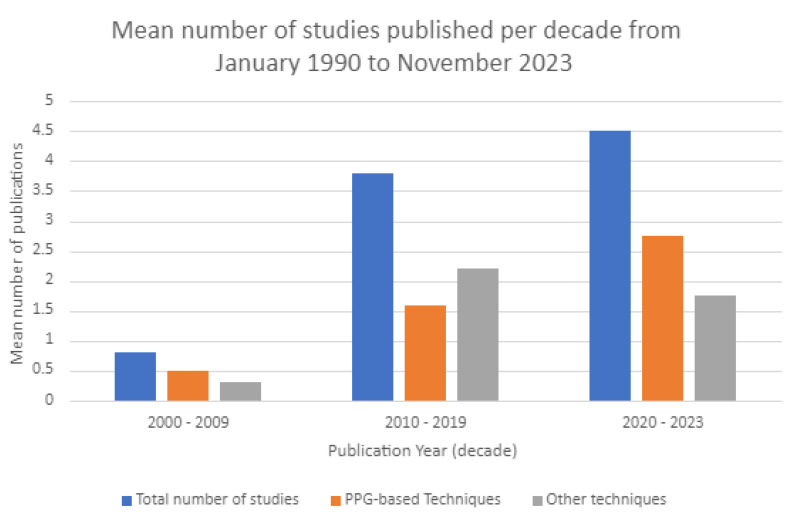
Bar chart showing the mean number of publications released per decade for the years between 2000 and 2023 that met the inclusion criteria. The total number of publications has been split between PPG-derived methodologies and other techniques used. No relevant studies were found for the years between 1990 and 1999.

**Figure 4 sensors-23-09882-f004:**
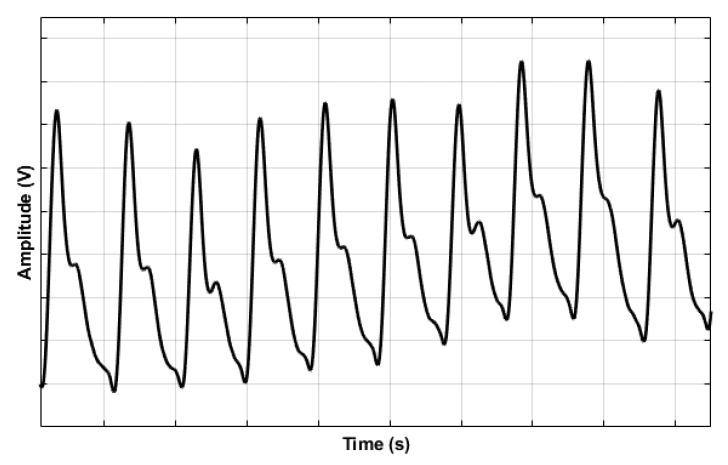
An example photoplethysmogram depicting the systolic and diastolic peaks with a dicrotic notch [45].

**Figure 5 sensors-23-09882-f005:**
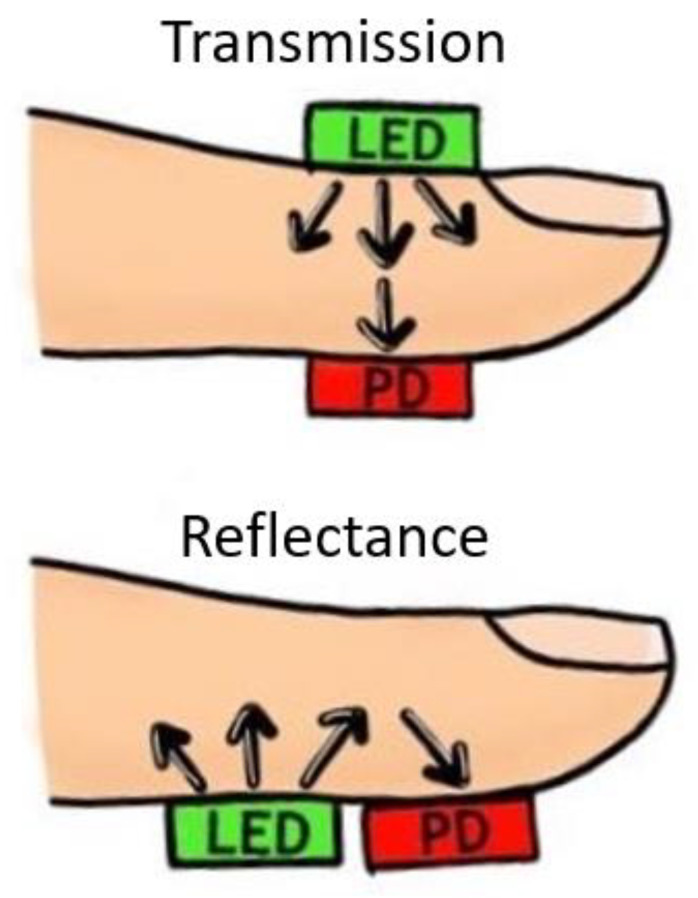
PPG sensor configuration modes. Transmittance mode configuration (**top image**) shows light source (LED) and photodetector (PD) placed on opposite sides of the body site. In reflectance mode (**bottom image**), the light source (LED) and photodetector (PD) are adjacent.

**Table 1 sensors-23-09882-t001:** Studies that used state-of-the-art techniques and technologies to assess vascular ageing.

Study	Year	Device(s)/Techniques	Measurement(s)	Number of Subjects	Major Findings
Gong et al. [10]	2023	Complior	Carotid femoral pulse wave velocity (cfPWV)	2063	In women over 60 years, body fat percentage (BFP) and cfPWV were correlated. There was no correlation detected for men or women under the age of 60.
Ji et al. [11]	2018	Not stated	cfPWV	Not stated	The method was able to successfully assess aortic pulse wave velocity (PWV) and measure central blood pressure (BP).
Cantürk et al. [12]	2017	Not stated	PWV	38	The mean PWV improved in 20 individuals, as indicated by an absolute decrease in PWV detected 6 months after having aortic valve replacement compared to a baseline, and deteriorated in the other 18 subjects. Generally found that aortic valve replacement had no effect on PWV mean and that aortic stenosis was related to baseline PWV.
Lurz et al. [13]	2017	Vicorder device	cfPWV	25	All subjects were found to have a normal cfPWV, suggesting that cfPWV does not change in subjects under 18 years of age.
Liu et al. [14]	2011	Computed tomography angiography	Brachial-ankle PWV (baPWV) Coronary Artery Calcium (CAC) score	654	In 127 patients, there was at least one coronary artery that was stenotic. In comparison to the normal control group, the stenotic group’s mean baPWV and mean CAC were considerably greater. The study found a link between coronary atherosclerosis and baPWV.
Tsuchikura et al. [15]	2010	Automated device	PWV	2806	In comparison to heart-carotid PWV (hcPWV), heart-brachial PWV (hbPWV), and femoral-ankle PWV (faPWV), it was discovered that baPWV had the strongest association with heart-femoral PWV (hfPWV). According to the findings, baPWV exhibited central stiffness as opposed to peripheral arterial stiffness.
Yufu et al. [16]	2004	Not stated	Aortic pulse wave analysis (PWA) Percent mean pulse amplitude (%MPA)	33	Higher arterial stiffness was indicated by a low %MPA. The %MPA was suggested as a novel atherosclerosis marker by the authors. It was discovered that the %MPA was lower in subjects with coronary artery stenosis than in those without it. The findings showed that brachial %MPA offered prognostic values for coronary atherosclerosis in people at risk for cardiovascular disease (CVD).
Scudder et al. [17]	2021	Dual impedance cardiography	PWV Pulse transit time (PTT)	78	Measurements of PWV showed a considerable positive connection with advancing age.
Miyoshi and Ito [18]	2016	Not stated	Cardio-ankle vascular index (CAVI)	Not stated	The study found that CAVI allows for the quantitative evaluation of disease progression, with higher CAVI values indicating a worse prognosis compared to lower CAVI values. CAVI was independent of BP at the time of measurement (unlike PWV) and indicated the overall stiffness of the artery from the aortic origin to the ankle. The study concluded that CAVI was a better identifier of disease severity than baPWV.
Souza-Neto et al. [19]	2016	Not stated	Ambulatory arterial stiffness index	85	The study found that those with a risk factor such as hypertension would have arterial stiffness. It was shown that arterial stiffness was substantially correlated with diabetes, hypertension, peripheral arterial disease (PAD), and coronary artery disease (CAD). Gender did not appear to impact arterial stiffness. Contrary to popular belief, age did not seem to be associated with arterial stiffness.
Klocke et al. [20]	2003	Not stated	Augmentation index (AIx)	14	The findings suggested that rheumatoid arthritis (RA) and increased arterial stiffness were related.
Zhang et al. [21]	2017	Oscillometric device	Arterial velocity–pulse index (AVI) Arterial pressure–volume index (API) BaPWV	183	It was discovered that baPWV, which was measured separately, was unable to foretell the existence of early atherosclerosis. BaPWV was found to be comparable to AVI in predicting CAD.
Wang et al. [23]	2011	Not stated	Compliance index (CI)	232	Measurements were made for both CI derived from digital volume pulse (DVP) and PWV-DVP; chronic kidney disease (CKD) patients had lower CI-DVP and greater PWV-DVP values than the healthy control group. Additionally, it was discovered that patients with late-stage CKD had lower CI-DVP levels than those with early-stage CKD. The more cardiovascular risk factors there were, the lower the CI-DVP was, according to the data.
Choy et al. [24]	2010	Oscillometric automated digital BP instrument	Arterial stiffness index (ASI) Ankle-brachial index (ABI) Arterial wave pattern	895	The risk of stroke was six times higher when the ASI was abnormal. Both ABI and arterial wave pattern showed strong correlations with stroke. However, there was a synergistic impact when assessing the risk of stroke when all three parameters were considered.
Naessen et al. [25]	2023	Ultrasound	Common carotid artery Intima thickness Intima/media (I/M) thickness ratio Intima–media thickness	63	It was discovered that compared to healthy subjects, patients with pulmonary arterial hypertension had an intima that was 56% thicker and an I/M ratio that was 128% higher. Patients with pulmonary arterial hypertension showed a thicker intima and greater I/M ratios than those with left ventricular heart failure with a reduced ejection fraction.
Li et al. [26]	2016	Real-time shear wave elastography	Longitudinal elasticity modulus PWV	179	PWV and systolic and diastolic BP were observed to be higher in AIS patients than in the control group. It was demonstrated that shear wave elastography could accurately and non-invasively measure the arterial wall’s longitudinal elastic modulus and assess arterial stiffness.
Bjällmark et al. [27]	2010	Conventional and Ultrasonographic strain measures	Common carotid artery elasticity	20	It was concluded that 2D strain imaging through ultrasonography was proven superior to traditional vascular stiffness measurements for determining the elastic characteristics of the common carotid artery.
Kang et al. [28]	2011	Cardiac magnetic resonance imaging (MRI)	Pulmonary artery distensibility index	35	The study aimed to determine whether the pulmonary artery stiffness estimated based on right heart catheterisation and Cardiac MRI-derived pulmonary artery distensibility corresponded.
Ha et al. [29]	2018	4D flow MRI	Aorta’s turbulent kinetic energy (TKE)	42	The study examined the extent and degree of turbulent blood flow in a healthy aorta and determined whether age has an impact on the turbulence level. TKE was a measurement of turbulence intensity. All the healthy subjects experienced turbulent flow in the aorta, and both groups’ aortas were similar overall. However, when compared to the younger participants, the older subjects had 73% greater total TKE in the ascending aorta. This was associated with age-related dilation of the ascending aorta, which increases the volume available for the generation of turbulence. It was determined that age-related geometric changes influenced the development of turbulent blood flow in the aortas of healthy subjects.
Sorelli et al. [30]	2018	Periflux 5000 Laser doppler flowmetry (LDF) system	Peripheral pulse Microvascular perfusion	54	A multi-Gaussian decomposition approach was applied to the LDF signals, and the algorithm proved effective at reconstructing the shape of the LDF pulses.
Rogers et al. [31]	2023	Near-infrared spectroscopy (NIRS) vascular occlusion testing	Age-related metabolic and microvascular function changes	34	It was concluded that the microvascular hyperaemic response and skeletal muscle metabolism decline with age for women.
Silva et al. [32]	2021	Mobil-O-Graph 24 h PWA Monitor Device Dual energy X-ray absorptiometry	Body composition PWV AIx Pulse Pressure Amplification Index Central Pulse Pressure	124	It was determined that arterial stiffness in the elderly is directly correlated with BFP. The findings could be related to an increased risk of cardiovascular disease.
Perrault et al. [33]	2019	SphygmoCor-Px System VP-1000 system EndoPAT 2000 system	PWV AIx Reactive hyperemia index (RHI)	40	A comparison was made between the three devices. A high level of PWV reliability was attained for both the VP-100 and SphygmoCor, as shown by a low coefficient of variation. AIx had a larger coefficient of variation when using the SphygmoCor or EndoPAT than PWV. The lack of association between RHI and AIx suggests that endothelial and artery parameters have different functional properties.
Markakis et al. [34]	2021	SphygmoCor-Px System Mobil-O-Graph 24 h PWA Monitor Device	Peripheral BP Central BP PWV Artificial Intelligence (AI)	57	Both devices were compared with patients experiencing hemodynamic shock in intensive care units (ICUs). The results showed that a lack of extra-vascular diseases made invasive procedures more reliable. However, the authors concluded that non-invasive techniques are practical and can be employed as extra monitoring techniques for shock patients. With the Mobil-O-Graph 24 h PWA Monitor Device and the SphygmoCor, full haemodynamic evaluations were successful in 48 patients and 29 patients, respectively. However, across the two devices, variations in the PWA were found.
Costa et al. [35]	2019	SphygmoCor-Px System	PWV	151	The PWV mean was observed to be higher in hypertensive patients. This finding suggests a clear relationship between arterial stiffness and increased PWV in the presence of hypertension. It was also discovered that arterial stiffness was more common in males who were older and had more risk factors than females.
Sridhar et al. [36]	2007	PeriScope	cfPWV baPWV	3969	Patients with RA had the highest heart rates whereas those with end-stage renal disease (ESRD) had the highest systolic BP. The PWV was discovered to be higher among people at higher risk of developing CVD, including those with CAD, diabetes mellitus, ESRD, and RA, compared to the healthy control group.
Komine et al. [37]	2012	Oscillometric BP device Form PWV/ABI vascular diagnostic device SonoSite 180 Plus	BP baPWV cfPWV Carotid arterial compliance	173	The study demonstrated that arterial stiffness can be assessed solely through cuff pressure oscillometric BP measurement. Similar to cfPWV and carotid arterial compliance, the estimated API had repeatability.
Hoffmann et al. [38]	2019	Oscillometric BP monitor	Heart rate Peripheral BP Central BP PWV	8	Long-term space flight’s vascular ageing biomarkers were assessed at baseline, 4 days, and 8 days post a 6-month International Space Station mission. Heart rate rose significantly 4 days after the return, but not on day 8, in comparison to the baseline. Additionally, central systolic BP also increased 4 days post-return versus the baseline measurement. PWV had an insignificant increase from baseline 4 days post-return and remained elevated on day 8. Overall, no clinically significant changes in early vascular ageing biomarkers were found in the evaluated cosmonauts.
Juganaru et al. [39]	2019	Anteriograph instrument	Waist-to-hip ratio	313	The device was able to identify patients at cardiovascular risk before any clinical indicators found arterial stiffness.
Osman et al. [40]	2017	Anteriograph instrument NICOM	Arterial stiffness PWV AIx Cardiac output Stroke volume Total peripheral resistance	33	Ultrasound scans were taken at five gestational windows between 11 and 40 weeks of pregnancy. It was discovered that normal pregnancy is associated with significant alterations in the maternal cardiovascular system. Arterial stiffness changes were observed in all measurements during healthy pregnancy, and the aortic PWV showed a significant variation during pregnancy.
Kostis et al. [41]	2021	Computational algorithm	Arterial stiffness 1 (AS1) Arterial stiffness 2 (AS2)	14097	Both indices were able to predict the occurrence of strokes. The study found that the indices derived from pulse pressure were more accurate at predicting the occurrence of stroke than pulse pressure or chronological age alone.
Negoita et al. [42]	2018	Computational algorithm: semi-automatic vendor-independent software Vivid E95 ultrasound	PWV	12	The study successfully developed software to trace the luminal diameter and blood velocity in the human ascending aorta by drawing from ultrasound images. The technique could calculate the PWV in the ascending aortas of adults.

**Table 2 sensors-23-09882-t002:** PPG-based studies for the assessment of arterial stiffness.

Study	Year	Device(s)/Technique	Parameters Measured	Number of Subjects	Major Findings
Allen and Murray [49]	2000	Bilateral photoplethysmography (PPG) study	Waveform of the pulses from six peripheral sites (including ears, thumbs, and big toes).	40	The system validation data showed low levels of root mean square error (RMSE), indicating good right-to-left channel matching. According to the bilateral study, normal patients’ pulses from their right and left sides were highly correlated at every segmental level (that is, the ears, thumbs, and toes).
Bentham et al. [50]	2018	Bilateral multi-site finger and toe PPG study	Multi-site finger and toe PPG	74	It was shown, using time-domain analysis, that those with PAD had lower normalised amplitude variability and significantly higher pulse arrival time (PAT) variability at the toe sites. In the frequency domain analysis, patients with PAD had noticeably decreased magnitude-squared coherence (MSC) values across a variety of frequency bands. It was discovered that the right toe had a different signal from the left toe while comparing the left and right sides of the body for the ear, finger, and toe. This resulted in the conclusion that PAD was at least present in one leg.
Brillante et al. [51]	2008	PPG	Peak to peak Stiffness index (SI) Reflection index (RI)	152	Age was found to significantly correlate with SI and RI, with race serving as an independent predictor of SI. At the same age, it was discovered that men had higher BP than women. There were no discernible differences between men and women in any of the arterial stiffness measures.
Jannasz et al. [52]	2023	SphygmoCor XCEL (AtCor Medina, Sydney, Australia)	Central and regional PWV cfPWV	118	cfPWV was discovered to be more reliable than regional PWV in determining arterial stiffness. There was insufficient evidence that gender affected the risk factors for atherosclerosis.
Tapolska et al. [53]	2019	Pulse Trace PCA 2 device	SI RI	295	Patients between the ages of 40 and 54 showed the greatest benefit from SI. The authors concluded that since SI and RI can be assessed non-invasively, they have the potential to be used in routine clinical practice to identify individuals who are at risk of developing future cardiovascular problems. Though RI can still be used as a supplementary measurement, it was found that it seemed to be less useful than SI.
Tanaka et al. [54]	2011	PPG	Finger arterial elasticity index (FEI) Finger arterial stiffness index (FSI)	199	The indices were gathered by occluding the finger, and future measurements of the stiffness of the small artery and arteriole can be made using the indices.
Wowern et al. [55]	2015	SphygmoCor-Px System Meridian digital pulse wave analysis (DPA)	PPG signals DPA Ejection Elasticity Index (EEI) Dicrotic Index (DI) Dicrotic Dilation Index (DDI)	112	It was discovered that the EEI should be used for large artery stiffness estimation while the DI and DDI should be used for small artery stiffness estimation.
Djurić et al. [56]	2023	PPG sensor	Blood flow Scalar coefficients	117	The study allowed for the analysis of amplitude changes in blood pulse waves. The scalar coefficient ratios declined with age, distinguishing between those above and below 50 years of age.
Huotari et al. [57]	2009	Transmission probe-based PPG device	Pulse waveforms from the left index finger and second toe	Not stated	It was discovered that PPG waveform analysis, as opposed to ultrasound analysis, offered greater details about artery structure and function.
Zekavat et al. [58]	2019	Finger PPG-derived ASI	ASI PPG BP	Approximately 500,000	It was determined that PPG-derived ASI was an inappropriate proxy for CAD risk but a genetically casual risk factor for BP.
Sharkey et al. [59]	2018	Bilateral PPG setup	Electrocardiogram (ECG) PPG PAT	181	A reduced PAT and thus higher arterial stiffness were discovered in heart transplant patients.
Wei et al. [60]	2013	PPG Radial pulse	cfPWV Spring constants	70	The PPG-based spring constant can distinguish between normal and pathological characteristics in both non-diabetic and diabetic individuals.
Park [61]	2023	SA-3000P (Medicore Co., Seoul, Republic of Korea) VS-1000 (Fukuda Denshi, Tokyo, Japan)	Second derivative of PPG signals CAVI	276	The second derivative of PPG, requiring a single transducer, proved simpler and yielded results in under 2 min. It was concluded that CAVI could not replace the second derivative of PPG.
Bortolotto et al. [62]	2000	Complior (Colson, Garges les Gonesses, France) Fukuda FCP-3166 (Fukuda, Tokyo, Japan)	cfPWV Second derivative of PPG signals	664	In patients over 60 with atherosclerosis, PWV and the second derivative of PPG remained higher, whereas in patients aged 60 with atherosclerosis, only PWV remained greater. While the index of the second derivative of PPG was correlated with age and other atherosclerosis risk factors, PWV was associated with age and arterial hypertension. The study found that aortic PWV more accurately captured changes in arterial compliance caused by ageing, high BP, and atherosclerosis than the second derivative of PPG.
Korneeva and Drapkina [63]	2015	PPG-based device	SI RI AIx Systolic BP	82	Obese patients with high BP showed vascular stiffness; based on pulse wave characteristics of the PPG signal, both treatments improved the arterial stiffness parameters, reducing SI and RI. Nonetheless, it was noted that compared to atorvastatin, rosuvastatin reduced the AIx.
Drapkina and Ivashkin [64]	2014	PPG-based device	SI RI AIx Systolic BP	82	The study found increased arterial stiffness in obese patients with high BP according to pulse wave analysis.
Drapkina [65]	2014	PPG-based device	SI RI AIx Systolic BP	82	The study found increased arterial stiffness in high-risk patients with high BP.
Wowern et al. [66]	2019	PPG	DPA	139	It was concluded that DPA reflects longitudinal changes in arterial compliance in normal pregnancy. In uncomplicated pregnancy, it was found that arterial stiffness changes in both large and small arteries significantly with gestational age.
Lefferts et al. [67]	2021	PPG Tonometry Ultrasound Doppler	BP cfPWV Carotid stiffness Blood velocity pulsatility index	30	Despite younger adults having larger carotid dilations, pulsatile damping decreased in both middle-aged and young adult groups. According to the study’s findings, the carotid diameter and cerebrovascular pulsatility are altered differently in young and middle-aged adults. The study suggested that changes in intracranial cerebral pulsatility could be slowed down by cerebrovascular characteristics.
Salvi et al. [68]	2008	Complior PulsePen PulseTrace	Aortic PWV SI	50	Complior and PulsePen were found to be accurate at estimating PWV via Bland–Altman analysis whereas PulseTrace was found to be an unsuitable substitute for PWV.
Djeldjli et al. [69]	2021	Imaging PPG (iPPG) Contact PPG (cPPG)	16 features of the pulse wave relating to arterial stiffness and BP	12	A comparison was made between the iPPG and cPPG, whereby cPPG recorded signals using contact probes and iPPG used a quick camera to record signals remotely. High agreement between the results and the features captured by the reference sensors was observed. The contact and contactless PPG features (from two separate sites) were found to share strong correlations. The non-invasive iPPG approach provided quantitative data on the underlying mechanisms of waveform shape from various body sites and allowed for a remote means to evaluate waveform features.
Kock et al. [70]	2019	ABI PPG	Peak to peak time	93	The results showed that ABI was not related to PPG indicators.
Murakami et al. [71]	2019	Rossmax International LTD SB200 pulse oximeter (Taiwan, China)	ASI baPWV	18	The fingertip PPG device was found to be capable of measuring arterial stiffness in line with the baPWV.
Clarenbach et al. [72]	2012	PPG Radial tonometry	SI AIx	83	Both devices distinguished between people with high and low cardiovascular risk accurately. However, unlike AIx, SI could also distinguish between people at intermediate and high risk, making it potentially more effective in sizable clinical research. Low failure rates were present in both devices.
Dall’Olio et al. [73]	2020	Machine learning (ML) Deep learning (DL)	Raw PPG signal	4769	It was discovered that individuals do not age at the same rate. The study reported that women had better health than men in terms of vascular ageing.
Shin et al. [74]	2022	DL	PPG pulse signals	752	The model outperformed earlier models at the time without the requirement for an additional feature detection process.
Park and Shin [75]	2022	Artificial neural network-based regression model	Incident and reflected waves from PPG waveform	757	It was determined that the reflected wave’s features may be used to assess vascular ageing, with the amplitude-related feature of the reflected wave being more favourable than the time-related feature in doing so.
Lian et al. [78]	2023	3 Lead ADS1292 ECG (Texas Instruments, Dallas, TX, USA) Remote PPG (rPPG)	ECG Heart rate	30	ECG signals were obtained as reference values. Video recordings from the rPPG were transferred for analysis; however, the results were limited to the Han ethnicity, lacking diversity.
Zhao et al. [79]	2023	700-MAX-ECG MONITOR (Maxim Integrated, San Jose, CA, USA) rPPG	ECG Heart rate	12	Unlike PPG and ECG, which may cause skin irritation and discomfort, rPPG offers a solution by eliminating adherence to the body.
Casalino et al. [80]	2023	rPPG	Blood oxygen saturation (SpO2)	10	Experimental results showed no significant differences with slight head movements. Further work is needed to explore potential light-condition impacts on real-time measurements.
Perpetuini et al. [81]	2019	Multi-site ECG and PPG	baPWV	78	It was discovered that the results were age-sensitive and had high signal quality since cross-talk effects were absent. The baPWV was found to be consistent when the results were compared to those from a commercial pulse sensor device, Enverdis Vascular Explorer (Düsseldorf, Germany).
Hydren et al. [83]	2019	Finger PPG-based device (Finapres Medical Systems, Amsterdam, The Netherlands) Doppler ultrasound	BP Single continuous pass leg movement (sPLM) Leg blood flow Leg vascular conductance	24	It was found that age-related decline in peripheral vascular function was found to be susceptible to sPLM. It was concluded that sPLM simplified the process and reduced the amount of equipment needed to just perform a Doppler ultrasound as compared to a standard passive leg movement.

## Data Availability

Data sharing not applicable.

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
