# Peer review of "Photoplethysmography for the Assessment of Arterial Stiffness"

_sensors, 2023, doi:10.3390/s23249882_

Round 1

Reviewer 1 Report

Comments and Suggestions for Authors

It is recommended to cite in the Vancouver format.

The bibliographic references in lines 53, 97, 107, 224, 289, 339 and 440 are incomplete.

Author Response

Reviewer 1

It is recommended to cite in the Vancouver format.

The bibliographic references in lines 53, 97, 107, 224, 289, 339 and 440 are incomplete.

Many thanks for your feedback. The references have been double checked and cited according to the submission guidelines.

Reviewer 2 Report

Comments and Suggestions for Authors

The authors propose a systematic review, based on the PRISMA protocol, of PPG techniques for vascular aging.

Overall, the following, the strengths and weaknesses of the paper are summarized:

Strengths:

 (+) The problem is important.

 (+) The problem is well-motivated.

 (+) The proposed method is well-explained.

Weaknesses:

 (-) There are English issues.

 (-) References are inadequate.

 (-) The introduction must be improved.

 (-) The related work section must be enhanced.

 (-) Experimental evaluation must be improved.

 (-) Some improvements are needed in the description of the method.

In the following more detailed comments about the manuscript:

==== REFERENCES ==== 

The literature review is incomplete. Several relevant references are missing. Particularly, the authors didn't mention the new findings on PPG consisting of remote PPG. This quite recent methodology avoids the need for contact, opening to a variety of applications, such as https://doi.org/10.1109/TIM.2023.3264041 https://doi.org/10.1007/s10489-023-04926-5  https://doi.org/10.3390/bios12020073

Particularly, in this recent work, computational models are combined with rPPG for vital sign parameters monitoring https://doi.org/10.1007/s12652-021-03635-6

==== INTRODUCTION ==== 

The introduction should clearly explain the key limitations of prior work (reviews) relevant to this paper.

Contributions should be highlighted more. It should be made clear what is novel and how it addresses the limitations of prior work. 

==== METHOD ==== 

It is important to explain better the design decisions (e.g. why the solution is designed like that). PRISMA protocol must be better explained and referenced.

==== EXPERIMENTS ==== 

There is not enough discussion of the experimental results. Graphs, summarizing the review findings would help the understanding of the research (e.g. a graph summarizing the devices used in the analyzed works).

Comments on the Quality of English Language

==== ENGLISH ==== 

The writing style of this paper is not good enough. Authors should spend some time improving it so that the paper can be read more smoothly.

The paper has several typos and missing references. Authors need to proofread the paper to eliminate all of them.

Author Response

Reviewer 2

The authors propose a systematic review, based on the PRISMA protocol, of PPG techniques for vascular aging.

REFERENCES  

The literature review is incomplete. Several relevant references are missing. Particularly, the authors didn't mention the new findings on PPG consisting of remote PPG. This quite recent methodology avoids the need for contact, opening to a variety of applications, such as https://doi.org/10.1109/TIM.2023.3264041 https://doi.org/10.1007/s10489-023-04926-5  https://doi.org/10.3390/bios12020073

Particularly, in this recent work, computational models are combined with rPPG for vital sign parameters monitoring https://doi.org/10.1007/s12652-021-03635-6

The authors would like to thank the reviewer for their recommendation on the relevant manuscripts. Some of manuscripts were published after the manuscript was originally written. The references suggested, as well as other recently published manuscripts, have been now reviewed and included into this manuscript. Also, the PRISMA flow diagram has been updated accordingly.

INTRODUCTION

The introduction should clearly explain the key limitations of prior work (reviews) relevant to this paper. Contributions should be highlighted more. It should be made clear what is novel and how it addresses the limitations of prior work. 

The authors would like to thank the reviewer for this comment. All the above recommendations have now been addressed in the introductory sections of the manuscript.

METHOD

It is important to explain better the design decisions (e.g. why the solution is designed like that). PRISMA protocol must be better explained and referenced.

We would like to thank the reviewer for this valid point. This has been addressed by citing and better explaining the PRISMA protocol in the caption of Figure 2.

EXPERIMENTS

There is not enough discussion of the experimental results. Graphs, summarizing the review findings would help the understanding of the research (e.g. a graph summarizing the devices used in the analyzed works).

We thank the reviewer for his valuable comment. We assume the reviewer is referring to experimental results of the literature covered in this review paper. Table 1 has be reviewed to make sure that all the results/major finding of all the pertinent current state of the art is captured correctly.

ENGLISH 

The writing style of this paper is not good enough. Authors should spend some time improving it so that the paper can be read more smoothly.

The paper has several typos and missing references. Authors need to proofread the paper to eliminate all of them.

The authors express their gratitude to the reviewer for highlighting this matter. The references have been thoroughly checked and adjusted in accordance with the submission guidelines. Additionally, the absent references have been reviewed and included. Two native English-speaking colleagues have proofread the paper.

Reviewer 3 Report

Comments and Suggestions for Authors

Dear Editors, Dear Authors,

Kyriacou's publication is a work that addresses a fundamentally relevant topic. In terms of content, it would be desirable for it to be made even clearer that this is a narrative review. It is regrettable that some of the references in the current version are not comprehensible. This should be addressed as a matter of urgency. The individual tables as well as the flow chart for the sources are very transparent and valuable. From my point of view, this work should be accepted in relation to the publication request for revision of the sources.

Author Response

Reviewer 3

Dear Editors, Dear Authors,

Kyriacou's publication is a work that addresses a fundamentally relevant topic. In terms of content, it would be desirable for it to be made even clearer that this is a narrative review. It is regrettable that some of the references in the current version are not comprehensible. This should be addressed as a matter of urgency. The individual tables as well as the flow chart for the sources are very transparent and valuable. From my point of view, this work should be accepted in relation to the publication request for revision of the sources.

The authors would like to thank the reviewer for their valuable suggestions and comments. We have amended the abstract and introduction accordingly to make it clear that this is a narrative review. We have also double checked the references and adjusted according to the submission guidelines. Some manuscripts have been published after this manuscript was initially written. These have been reviewed and included in the manuscript.  

Reviewer 4 Report

Comments and Suggestions for Authors

Karimpour et al submitted this review manuscript that focused on assessment of arterial stiffnesses. The review discusses the current techniques that are being used in measurement of vascular arterial stiffness with highlighting the importance of using Photoplethysmography. Overall, the review is written well, which may provide a better understanding of these methods to find a better approach. However, this reviewer has some comments (below) that need to be addressed.

-       Along with structural changes, (e.g., arterial stiffness) vascular aging can also cause other changes in the arteries (e.g., expressions and functions of signaling molecules). Thus, the title of the review can be more specific to the measurement of the arterial stiffness.

 -        Although mentioned in the text that there has been increase in the number of published articles on PPG, Figure 3 shows a trend for decrease in PPG-based Techniques during 2020-2023. Please explain that.

 -        Please provide details about the figures 1, 2 & 4 in their figure legends.

 -        It would be helpful if a schematic/illustration of how PPG works is added to the manuscript. It can be combined with some recorded representative raw traces as in figure 4.  

 -        Including a separate section on limitations of PPG method would be good. It can be under section 4.

Author Response

Review 4

Karimpour et al submitted this review manuscript that focused on assessment of arterial stiffnesses. The review discusses the current techniques that are being used in measurement of vascular arterial stiffness with highlighting the importance of using Photoplethysmography. Overall, the review is written well, which may provide a better understanding of these methods to find a better approach. However, this reviewer has some comments (below) that need to be addressed.

Along with structural changes, (e.g., arterial stiffness) vascular aging can also cause other changes in the arteries (e.g., expressions and functions of signaling molecules). Thus, the title of the review can be more specific to the measurement of the arterial stiffness.

Many thanks for the comment. The authors of the paper have made the title more specific to arterial stiffness.

Although mentioned in the text that there has been increase in the number of published articles on PPG, Figure 3 shows a trend for decrease in PPG-based Techniques during 2020-2023. Please explain that.

To better show the trend, the authors have changed Figure 3 to show the mean number of publications per decade. This way, it is clearer that the trend shows an increase in the number of articles on PPG. Furthermore, it should be noted that from July 2023 (when the manuscript was originally written) until now, more papers have been published. These have been reviewed and included into the manuscript and is reflected in Figure 3.

Please provide details about the figures 1, 2 & 4 in their figure legends.

Thank you for this valuable comment. More detail has been provided for Figures 1, 2 and 4 in their figure captions to better explain the figures.

It would be helpful if a schematic/illustration of how PPG works is added to the manuscript. It can be combined with some recorded representative raw traces as in figure 4.  

Many thanks for your feedback. In section 4 we have included an illustration (Figure 5) to show how PPG works in both transmittance and reflectance modes.

Including a separate section on limitations of PPG method would be good. It can be under section 4.

Thank you for the advice. The authors have implemented a separate paragraph on the limitations of using PPG at the end of section 4.

Round 2

Reviewer 2 Report

Comments and Suggestions for Authors

Authors have properly enriched their work, by addressing each comment in a suitable way. The paper turns out to be notably improved.